# The Mechanism of Modulation of Cardiac Force by Temperature

**DOI:** 10.3390/ijms26020469

**Published:** 2025-01-08

**Authors:** Ilaria Morotti, Matteo Marcello, Giulia Sautariello, Irene Pertici, Pasquale Bianco, Gabriella Piazzesi, Marco Linari, Vincenzo Lombardi, Massimo Reconditi, Marco Caremani

**Affiliations:** 1PhysioLab, University of Florence, 50019 Sesto Fiorentino, Italy; ilaria.morotti@unifi.it (I.M.); mmarcello@genethon.fr (M.M.); giulia.sautariello@unifi.it (G.S.); irene.pertici@unifi.it (I.P.); pasquale.bianco@unifi.it (P.B.); gabriella.piazzesi@unifi.it (G.P.); marco.linari@unifi.it (M.L.); massimo.reconditi@unifi.it (M.R.); marco.caremani@unifi.it (M.C.); 2Department of Biology, University of Florence, 50019 Sesto Fiorentino, Italy; 3Department of Experimental and Clinical Medicine, University of Florence, 50134 Florence, Italy

**Keywords:** cardiac muscle mechanics, cardiac myosin, temperature effect on cardiac output, force generation by cardiac myosin

## Abstract

In maximally Ca^2+^-activated demembranated fibres from the mammalian skeletal muscle, the depression of the force by lowering the temperature below the physiological level (~35 °C) is explained by the reduction of force in the myosin motor. Instead, cooling is reported to not affect the force per motor in Ca^2+^-activated cardiac trabeculae from the rat ventricle. Here, the mechanism of the cardiac performance depression by cooling is reinvestigated with fast sarcomere-level mechanics. We determine the changes in the half-sarcomere compliance of maximally Ca^2+^-activated demembranated rat trabeculae in the range of temperatures of 10–30 °C and analyse the data in terms of a simplified mechanical model of the half-sarcomere to extract the contribution of myofilaments and myosin motors. We find that the changes in the ensemble force are due to changes in the force per motor, while the fraction of actin-attached motors remains constant independent of temperature. The results demonstrate that in the cardiac myosin, as in the skeletal muscle myosin, the force-generating transition is endothermic. The underlying large heat absorption indicates the interaction of extended hydrophobic surfaces within the myosin motor, like those suggested by the crystallographic model of the working stroke.

## 1. Introduction

Muscle contraction is due to ATP-driven interactions between myosin motors and actin filaments. In each sarcomere, the ~2 μm-long structural unit of the striated muscle (skeletal and cardiac), myosin motors, extending in two bipolar arrays from the thick filaments at the centre of the sarcomere, produce force and shortening by asynchronous cycles of attachment to and detachment from actin on the overlapping thin filaments, originating from the Z line bounding the sarcomere. Contraction is regulated by mechanisms on both thin and thick filaments. The first mechanism, based on Ca^2+^-dependent structural changes of the thin filament regulatory proteins, makes the actin sites available for binding of the myosin motors shortly after the action potential [1]. The second mechanism recruits the myosin motors from the OFF state, in which they lie on the surface of the thick filament folded on their tail in the so-called interacting heads motif (IHM) [2,3,4], unable to bind actin and hydrolyse ATP [5,6]. By using X-ray diffraction from intact preparations of skeletal and cardiac muscles, it was shown that in the thick filament a mechanosensing mechanism operates, which moves motors away from the surface of the filament in relation to the contraction load, making them available for interaction with actin [7,8,9,10].

In mammals, both skeletal and cardiac muscles are known to reduce their mechanical output with the decrease of temperature below the physiological value (35–37 °C). The mechanisms behind this phenomenon can be related to either a decreased activation of myofilaments, which in turn decreases the available number of actin–myosin interactions, or the decrease of the mechanical performance of the myosin motors themselves as a consequence of a direct effect of temperature on the mechanokinetic parameters of the actin–myosin interaction.

Mechanical and time-resolved X-ray diffraction experiments on skeletal muscle demonstrated that in heterothermic animals like the frog the reduction of the active isometric force with cooling can be solely accounted for by depression of the force-generating capability of myosin motors without a reduction in the number of actin-attached motors [11,12,13]. Meanwhile, in the intact fast muscle of the mouse, the much larger depression of isometric force by cooling is explained by the decrease in both number and force per motor [14]. Cooling of mouse muscle also produces a reduction in the number of motors in the OFF state at rest, suggesting that the reduced number of attached motors upon stimulation is due to accumulation of motors at rest in a disordered refractory state that makes them unavailable for interaction with actin.

In the heart, the mechanical activity (systole) consists of short periodic contractions (twitches) triggered by single action potentials, during which the blood is pumped by ventricles into the arterial circulation. In the resting period between two systoles (diastole), the heart is filled with blood from the venous return. The transient increase of calcium with the action potential does not reach the level for full thin filament activation, so the mechanical response depends on both the cytosolic concentration of Ca^2+^ ([Ca^2+^]_i_) and the Ca^2+^ sensitivity of the filament [15,16]. Also in the cardiac muscle in diastole, as in the resting skeletal muscle of the mouse, lowering the temperature reduces the number of motors in the OFF state [10,17], an effect that could indicate a rise in the population of motors in a disordered refractory state that makes them unavailable for interaction with actin.

A specific complexity in the regulation of heart performance and its dependence on temperature derives from the fact that [Ca^2+^]_i_ attained during the twitch and the time course of its decay depend on transporters, such as the Na^+^–Ca^2+^ exchanger and the sarcoplasmic reticulum Ca^2+^-ATPase [18,19], which are temperature sensitive. With an external concentration of Ca^2+^ ([Ca^2+^]_o_) of 1–2 mM, and a stimulation frequency of 0.5 Hz, the peak force of the twitch, *T*_p_, is maximum in the temperature range 25–30 °C and decreases approaching the physiological temperature [20,21], at which the Ca^2+^ transient is smaller and briefer [21]. The same explanation should apply to the finding that at temperatures below the physiological one, *T*_p_ reduces with the increase in the pacing frequency (22 °C, [21]; 27 °C, [10]).

Thus, the direct effect of cooling on the parameters characterising the thick filament contribution to the mechanical performance of the cardiac myocyte, such as the number of force-generating motors or the force per motor, cannot be directly inferred from the relation between the force of the cardiac twitch and temperature, as they are entangled with the effect of temperature on the calcium transient and the consequent calcium-dependent thin filament activation.

This problem can be overcome by determining the relation between isometric force and temperature in demembranated myocytes, in which the concentration of activating Ca^2+^ is set by the experimenter. Previous work in demembranated cardiac preparations from rabbits [22] and rats [23] reported that both the maximum isometric force (*T*_0_) developed at saturating [Ca^2+^] (~50 μM, pCa (=−log_10_[Ca^2+^]) = 4.3) and the Ca^2+^ sensitivity (measured by the pCa value at which the isometric force is ½ *T*_0_) decreased with the reduction of the temperature. These results and the temperature dependence of the relations between other mechanokinetic parameters and pCa were integrated by de Tombe and Stienen [23] in a simple two-state model [24] to conclude that only the absolute values of attachment and detachment rate constants were temperature dependent, while both the fraction of attached motors and the force per motor were almost unaffected by temperature. These conclusions pose two questions: the first is whether, in contrast to what was found in skeletal muscle myosin, the force-generating process in cardiac myosin is not temperature sensitive, and the second is whether the drop in isometric force of the cardiac preparation with cooling is attributable to the rise of a refractory state of the motors, as in mouse skeletal muscle.

Here, these questions are investigated with fast sarcomere-level mechanics in Ca^2+^-activated demembranated trabeculae dissected from the ventricle of a rat, by determining the dependence on temperature of the isometric force (*T*_0_) and of the underlying mechanical parameters at the molecular level, such as the number of attached motors and the force per motor. In this scope, we record the elastic response to step perturbations in length superimposed on the steady force developed during maximal Ca^2+^ activation in the range of temperatures of 10–30 °C. The results are analysed in terms of a simplified mechanical model of the half-sarcomere [25], to extract the contribution of myofilaments and myosin motors to the compliance of the half-sarcomere (hs) at each temperature. We find that the fraction of actin-attached motors that contribute to the force of the ensemble remains constant, independent of temperature, and that the drop in the ensemble force (*T*_0_) with lowering temperature is totally accounted for by the reduction of the force per motor and excludes the rise of a refractory state of the thick filament with cooling. The results demonstrate that in the cardiac muscle, as in the skeletal muscle, the force-generating transition in the myosin motor is an endothermic process characterised by similar thermodynamic parameters. The large heat absorption associated with the transition finds a likely explanation in the extended hydrophobic surfaces involved in the interdomain interactions within the myosin motor that, according to the crystallographic model [26], lead to the working stroke that drives filament sliding when the load is lower than the isometric force.

## 2. Results

### 2.1. Temperature Dependence of Force and Stiffness of the Half-Sarcomere in the Maximally Activated Demembranated Trabecula

Both the steady isometric force of maximally Ca^2+^-activated trabeculae (*T*_0_) and the rate of force redevelopment (*k*_TR_) following a large length release decreased with the decrease in temperature from the near-physiological value (30 °C). The relation between *T*_0_ and temperature (Figure 1A) showed an extended region of maximum force at near physiological temperature (25–30 °C; Table 1) and then a decrease at lower temperatures. At 10 °C, the *T*_0_ value was reduced by ~40%. Instead, the relation between *k*_TR_ and temperature showed a sharp continuous decrease with the decrease in temperature below 30 °C, attaining a *k*_TR_ value five-fold lower at 10 °C (Figure 1B,C and Table 1).

The molecular basis of the effect of temperature on the isometric force was investigated by measuring the hs stiffness, *k*_0_, at the different temperatures. The stiffness of the array of motors can be calculated from *k*_0_ by exploiting a simplified two-component mechanical model of the half-sarcomere.

As detailed in the Methods Section, *k*_0_ was measured by superimposing rapid length changes on the steady isometric force developed at saturating [Ca^2+^] (Figure 2 and Figure 3A).

The length step elicited a simultaneous change in force, which measured the elastic properties of the half-sarcomere, followed by a quick partial recovery of force due to the execution of the working stroke by the attached motors synchronised by the length step [27,28] (Figure 3A, left panel, 10 °C; right panel, 25 °C). The speed of the quick force recovery was larger for step releases than for step stretches, and for step releases it increased with the step amplitude. This would increase the degree of truncation of the force attained at the end of the step, *T*_1_ (arrowhead in Figure 3A), going from the largest stretch to the largest release. However, for small length changes completed within 100 μs, like those used here, the truncation was negligible, as was made evident by the inspection of the relation between *T*_1_ and the change in the half-sarcomere length (Figure 3B), which appeared linear, as expected from a purely elastic response. The slope of the *T*_1_ relation measures *k*_0_, while the intercept on the length axis measures the strain in the half-sarcomere (*Y*_0_) at the isometric force attained just before the length step. It is evident from Figure 3B that *k*_0_ was the same at 25 °C (dashed line, linear fit to the open symbols) as at 10 °C (continuous line, linear fit to the filled symbols). *k*_0_ from six experiments was 15.1 ± 0.4 kPa∙nm^−1^ at 30 °C and 15.5 ± 1.1 kPa∙nm^−1^ at 10 °C (Table 1). Instead, *Y*_0_ was smaller at 10 °C (4.8 ± 0.4 nm) than at 30 °C (8.5 ± 0.3 nm). The reduction was by 43%, the same as the reduction in isometric force. Consequently, while the relation of *k*_0_ versus the force developed at each temperature (Figure 3C) showed that the stiffness did not change with the temperature-dependent change in isometric force, the relation of *Y*_0_ versus the force developed at each temperature (Figure 3D) showed that the strain of the half-sarcomere decreased in proportion to the decrease in isometric force with lowering temperature.

### 2.2. Estimate of the Temperature Dependence of the Mechanical Parameters of the Motor Array Through the Half-Sarcomere Compliance Analysis

The results in the previous section were analysed on the basis of a simple two-component mechanical model of the half-sarcomere (Figure 4A), in which the elasticity is given by two elements in series: an element with constant stiffness, the myofilaments, and an element, made by the array of attached myosin motors, that changes its stiffness in proportion to the number of motors in the array [25,29,30,31].

For the analysis of the components of the half-sarcomere elasticity, it is convenient to use the half-sarcomere compliance, *C*_hs_, the reciprocal of *k*_0_, because *C*_hs_ (nm/MPa) is given by the sum of the contributions of its in-series components:*C*_hs_ = *C*_f_ + 1/(β·*e*_r_),(1)
where *C*_f_ is the equivalent myofilament compliance and 1/(β·*e*_r_) is the compliance of the array of myosin motors in the half-sarcomere, with *e*_r_ being the stiffness of the array when all the motors are attached, and β is the fraction of attached motors.

*C*_hs_ was constant in the whole range of the temperature-modulated *T*_0_ (Figure 4B). Therefore, given that the contribution of myofilament compliance is constant independent of temperature, the array of motors also contributes to *C*_hs_ with a compliance that is constant independent of the value of the temperature-modulated *T*_0_. This, in turn, indicates that the fraction of attached motors, β, does not change significantly with temperature. Here, 1/(β·*e*_r_) corresponds to *s*_0_/*T*_0_, where *s*_0_ is the average strain shared by the attached motors (as they are in parallel) contributing to the ensemble force *T*_0_. Thus, *C*_hs_ can also be expressed as follows:*C*_hs_ = *C*_f_ + *s*_0_/*T*_0_(2)

The myosin motor has a constant stiffness independent of the generated force [28,32], thus *s*_0_ in isometric contraction is purely related to the force attained by the force-generating transitions in the motor. From (2), *s*_0_ can be calculated as follows:*s*_0_ = (*C*_hs_ − *C*_f_)·*T*_0_(3)

*C*_f_, estimated in the same preparation by Pertici et al. [33], was 16.0 ± 2.2 nm/MPa, and *s*_0_, calculated with (3) for each *T*_0_, is reported in Figure 4C, left ordinate. It can be seen that *s*_0_ increased in proportion to *T*_0_. Knowing the stiffness of the motor (*ε*), the corresponding force per motor (*F*_0_) can be calculated as *F*_0_= *ε·s*_0_. The value of *ε* estimated from the stiffness of the half-sarcomere of the same preparation in rigor [34] was 1.07 pN/nm. Assuming that the stiffness of the motor is constant independent of temperature, the relation between *F*_0_ and the temperature-dependent *T*_0_ can be calculated (right ordinate in Figure 4C). *F*_0_ increased from 3.9 ± 0.4 pN to 6.9 ± 0.2 pN in the range of 10–30 °C. The direct proportionality between *F*_0_ and *T*_0_ led to the conclusion that temperature modulated the force of the motor ensemble by its effect on the force-generating capability of the motor at a constant fraction of attached motors.

## 3. Discussion

The main contribution of this work was the demonstration that in the cardiac muscle, as in the skeletal muscle, force generation by the myosin motor was an endothermic process, so the force of the motor increased with the increase in temperature and was maximum as the temperature approached the physiological value. The study implied a methodological approach that combined the definition of the effects of temperature on the active force and stiffness at the level of the half-sarcomere with the interpretation of data using a simple two-component model of the half-sarcomere. The conclusion of the work made evident a striking difference between the effects of temperature and those of other interventions able to modulate the response of the cardiac muscle, such as the change in concentration of the activating Ca^2+^. By using the same methodological approach, it was demonstrated that the relation between myoplasmic [Ca^2+^] and force was totally accounted for by the modulation of the number of attached myosin motors by Ca^2+^, while the force and the strain of the motor remained constant [33,34]. Notably, in those studies, the assumption of the two-component model of Figure 4A resulted in an excessive simplification, unable to fit the data at low forces unless it was integrated with a parallel elastic element with constant stiffness ~1/10 of that of the motor array at saturating [Ca^2+^] (see Figure 3A in Pinzauti et al. [34]). The additional parallel elastic element was likely titin and became evident only when the number of attached motors is so low to provide a comparable equivalent stiffness [25]. In these experiments, contractions were elicited at the saturating Ca^2+^ concentration, at which the number of attached motors stayed constantly high at each temperature. Consequently, at any force, titin stiffness was so low in comparison to the stiffness of the motor array that its contribution to the half-sarcomere stiffness was negligible, justifying the assumption of a simplified two-component model in these experiments.

From the skinned trabeculae studies, it is known that a concentration of Ca^2+^ of ~10^−6^ M, which is the value reached by myoplasmic [Ca^2+^] in the intact myocyte during the cardiac systole [35,36], is within the steep region of the relation between Ca^2+^ concentration and force [23,33]. This gives the molecular explanation of the dependence of the mechanical performance of the intact myocyte on modulatory interventions that change the internal [Ca^2+^] through their effect on the transporters [18,19]. On the other hand, the unsuitability of the intact trabecula for the investigation of the direct effect of temperature on the mechanical performance of the cardiac half-sarcomere became evident, as temperature also affected the kinetics of the transporters and thus of the Ca^2+^ transient.

In the skinned trabecula used in this work, the direct effect of temperature on the performance of the myosin motors was investigated with fast sarcomere-level mechanics during maximal Ca^2+^ activation. In this way, we established that temperature did not affect the number of motors, while it modulated the force per motor. The finding that the fraction of attached motors was not affected by temperature accords with similar results obtained in intact and skinned fibres from skeletal muscle [11,12,37,38,39,40]. Thus, the endothermic process that relates the isometric force of the active muscle to the temperature was the force-generating transition in the muscle myosin. At odds with this conclusion, Kawai and coworkers, in skinned fibres from rabbit psoas [41,42], found that the temperature-dependent increase in force was explained by the increase in the number of attached force-generating motors and excluded the increase of the force per motor. However, their stiffness estimate may have been influenced by a large series compliance (which was not taken into account) and a reduced time resolution of mechanical measurements (the number of attached heads was deduced from stiffness estimates extrapolated from tension responses to length oscillations at frequencies of 100 Hz, one order of magnitude lower than the frequency necessary for isolating the elastic response, corresponding to the 100 µs step applied in the measurements presented here).

Most of the studies on the effect of temperature on the isometric force included the analysis of the effects on the corresponding kinetic parameters, such as the rate of ATP hydrolysis of actomyosin (*k*_cat_), the velocity of unloaded shortening (*V*_max_), and the rate of force redevelopment following a large release (*k*_TR_). The *Q*_10_ (the ratio of the value of the parameter under investigation at a temperature over that at a temperature 10 °C lower) was found systematically higher for the kinetic parameters than for the isometric force. In particular, in the rat skinned trabecula, de Tombe and Stienen [23] found a *Q*_10_ of 1.4 for the force (the same as in this work, Table 1) and twice larger *Q*_10_ values for *k*_cat_ and *k*_TR_. Notably, a larger temperature sensitivity of the kinetic parameters can be appreciated in this work as far as *k*_TR_ (Figure 1C), the temperature dependence of which showed a *Q*_10_ of 2.2 (Table 1), comparable to that reported by de Tombe and Stienen [23].

In terms of the simple two-state model of the actin–myosin ATPase mechanochemical cycle [24], motor attachment–force generation is controlled by the rate constant *f* and motor detachment is controlled by the strain-sensitive rate constant *g*, which increased with the shortening as the position of the attached motors shifted towards negative values relative to the position of attachment on actin.

According to this scheme, the fraction of force-generating motors (the duty ratio *r*) is expressed by *f*/(*f* + *g*), *k*_cat_ is expressed by *f*·*g*/(*f* + *g*), and *k*_TR_ is expressed by *f* + *g*. Given the strain dependence of *g*, reducing the load of the contraction and thus increasing the shortening velocity, *r* is expected to decrease with the velocity of shortening, while *k*_cat_ is expected to increase up to a maximum that corresponds to *k*_cat_ in solution (*k*_cat,s_). The bulk of experimental data showing that *Q*_10_ was lower for the isometric force *T*_0_ than for the corresponding *k*_cat_ indicated that the tension cost, which is the ratio between *k*_cat_ at *T*_0_ and *T*_0_ (proportional to *r*), increased with temperature. This, in turn, indicated that *g* (=*k*_cat_/*r*), which is proportional to tension cost, increased with temperature. These conclusions of the biochemical–mechanical study of de Tombe and Stienen [23], integrated with the finding that *r* was constant independent of temperature, indicated that temperature increased to the same extent as the kinetic parameters *f* and *g*. Correspondingly, *k*_TR_ is expected to increase with the same *Q*_10_ as *k*_cat_. In summary, the work of de Tombe and Stienen [23], which missed a direct reliable measure of the effect of temperature on *r*, and our work, which missed the measure of the effects of temperature on *k*_cat_, converge to the same conclusion that the rise of kinetic parameters with temperature did not change *r*. The two works, however, disagree on the effect of temperature on *F*_0_: while the compliance analysis of the active half-sarcomere at different temperatures in this work showed that the temperature-dependent increase in *T*_0_ is explained by a correspondingly larger *F*_0_, de Tombe and Stienen [23], on the basis of their mechanokinetic parameters, calculated values of *F*_0_ that did not change significantly with temperature. This conclusion, however, carries the intrinsic contradiction that neither their estimate of *F*_0_ nor that of *r* (the parameters determining the force of the motor ensemble *T*_0_) accounted for the temperature-dependent changes in *T*_0_.

The finding in this work that the potentiating effect of temperature on *T*_0_ was totally accounted for by the increase in *F*_0_ allowed the identification of the endothermic nature of the force-generating transition in the myosin motor. Under this condition, *T*_0_ data in Figure 1A were replotted as a function of the reciprocal of absolute temperature (1/Θ; Figure 5) and fitted with the equation (derived from the van ’t Hoff equation):*T*_0_ = *T*_0,max_/(1 + exp(Δ*H*/(*k*_b_·Θ) − Δ*S*/*k*_b_)),(4)
where *T*_0,max_ is the maximum isometric force attained with the rise in temperature, Δ*H* is the enthalpy change, *k*_b_ is the Boltzmann constant, Δ*S* is the entropy change, and Δ*H −* Θ·Δ*S* = Δ*G* is the change in free energy that in the isometric condition accounts for the mechanical energy stored in the elastic element of the motor as a consequence of the state transition. The fit of (4), continuous line, yielded *T*_0,max_ = 143 ± 7 kPa, Δ*H* = 199 ± 47 zJ, and Δ*S* = 0.7 ± 0.2 zJ K^−1^. Since Δ*H* and Δ*S* were independent of temperature, the rise in the entropic factor Θ·Δ*S* was responsible for the negative sign and size of Δ*G* and thus of the temperature-dependent mechanical energy stored with the transition.

The endothermic nature of force generation by the myosin motor associated with a large increase in entropy indicated that the hydrophobic amino acid interactions that reduced the extent of the cage of ordered water molecules around the hydrophobic residues were the main underlying molecular process [42]. The released water molecules were more disordered and thus entropy increased, and the free energy decreased. This characteristic of the force-generating transition can be used to identify the domains in the actin–myosin complex responsible for the force-generating transition. According to the crystallographic model [43], the actin–myosin interface that is responsible for the attachment of myosin to actin involves a surface area not sufficient to explain the large heat absorption consistent with the change in Δ*H* (∼200 zJ per motor) found here and in skeletal muscle [40,42,44]. Instead, the conformational changes in the myosin motor responsible for the working stroke (like the closure of the 50 kD cleft and the rotation of the converter) concern hydrophobic surface areas extended enough to account for the change in Δ*H*. This mechanism of force generation finds further support in previous work on intact fibres from skeletal muscle [12]. It was found that, following a stepwise drop in the otherwise maximum isometric force to a lower value, the early rapid shortening that represents the mechanical manifestation of the working stroke in the attached motors synchronised by the step [45] was smaller at higher temperatures, showing that the rise in isometric force with temperature occurred at the expense of the early steps of the same series of state transitions of the working stroke driving filament sliding.

In apparent contrast with this conclusion, it has been found that the rise of the force elicited with a stepwise increase in temperature (T-jump) was more than one order of magnitude slower than the early force recovery following a step release [37,39,46,47,48,49,50,51,52]. The simplest explanation of this finding was that the T-jump primarily targets the two endothermic steps that precede the force-generating transition in the myosin motor, namely, the ATP hydrolysis and the subsequent attachment to actin [53]. In that case, the temperature-dependent increase in force would be the consequence of the mass action that raises the occupancy of the higher force-generating state. This explanation, however, would imply a significant increase in the number of attached motors, which was not found, and, moreover, fails to consider the effect of the strain dependence of the rate constants controlling the force-generating transition at the high load at which the T-jump force transient is defined. This can be appreciated considering the isotonic velocity transients elicited with step perturbations in force [54]. The velocity of the early rapid shortening, the mechanical manifestation of the working stroke, was 12,000 nm s^−1^ following a drop of isometric force (*T*_0_) to 0.1 *T*_0_ and decreased by 15 times if the force was dropped to 0.8 *T*_0_. Note that the transient elicited by a step to a force 20% lower than the equilibrium force occurred under comparable loading conditions as the transient elicited by a T-jump imposed on the isometric force. The quick recovery from a step release, instead, occurred under continuously increasing loading conditions, starting from a very low one in correspondence with the end of the step. Consequently, the rate of the force transient following a T-jump, such as the rate of the early phase of shortening following a force drop to 0.8 *T*_0_, is expected to be more than ten-fold smaller than the rate of the force transient elicited by a step release. In fact, it has been demonstrated [55] that a unique mechanokinetic model with opportune assumptions for the strain dependency of the rate constants of the force-generating state transition is able to fit the experimental relations between either, in length clamp, the rate of quick force recovery and the size of the step release or, in force clamp, the rate of early shortening and the size of the force drop. We can conclude that, even if the increase in temperature also influences other transitions, such as the ATP hydrolysis and the subsequent attachment to actin, it selectively targets the force-generating transition, revealing its endothermic nature. These conclusions find further solid support in the present analysis of the compliance of the active half-sarcomere, demonstrating that the potentiating effect of temperature was accounted for by the increase in the force per motor, without a change in the fraction of actin-attached motors.

Sarcomere-level mechanics in demembranated trabeculae was used to study the effect of temperature on the force and stiffness of the active half-sarcomere and, in combination with a two-component equivalent model of the half-sarcomere, to define in situ the mechanokinetic and thermodynamical parameters of the force-generating mechanism in the myosin motor. The success of the approach exceeds the scope of this work, as it can also be applied to the investigation of the effects of cardiomyopathy-causing mutations in the contractile proteins on the force-generating mechanism of demembranated preparations from animal models and human biopsies. Defining the effect of mutations on the basic force-generating mechanism is the prerequisite for targeting it with potential therapeutic interventions.

## 4. Materials and Methods

### 4.1. Animals and Ethical Approval

Animals (male rats, *Rattus norvegicus*, strain Wistar Han, 230–280 g, aged 2–3 months) were treated in accordance with both the Italian regulation on animal experimentation (Authorisation No. 17E9C.N.CLU), in compliance with Decreto Legislativo 26/2014, and the EU regulation (Directive 2010/63). All animals were kept with free access to food and water prior to use. Experiments were performed in compliance with the principles of the 4 Rs. Animals were anaesthetised with isoflurane (5%, *v*/*v*). As soon as the animal was deeply anaesthetised, as judged by the absence of the pedal reflex and the loss of the muscle tone in the hindlimb, the heart was rapidly excised, placed in a dissection dish, and retrogradely perfused with a modified Krebs–Henseleit (K-H) solution (in mM: 115 NaCl, 4.7 KCl, 1.2 MgSO_4_, 1.2 KH_2_PO_4_, 25 NaHCO_3_, 0.5 CaCl_2_, and 10 glucose), containing 20 mM of 2,3-butanedione monoxime (BDM) and equilibrated with carbogen (95% O_2_, 5% CO_2_, pH 7.4). Chemicals were obtained from Sigma (St. Louis, MO, USA), carbogen from SAPIO (Monza, Italy).

### 4.2. Sample Preparation

From each heart, one unbranched and uniform trabecula was dissected from the right ventricle under a stereomicroscope. From a total of eight dissected hearts, two of them did not contribute to the analysis, as the shape and size of their trabeculae were not adequate to provide reliable sarcomere length change signals. Isolated trabeculae were transferred in a Petri dish with the bottom covered by a layer of Sylgard (Dow Corning, Midland, Michigan) and demembranated (skinned) by ~30 min perfusion at room temperature with 1% (*v*/*v*) Triton X-100 in skinning solution (Eastwood A, Table 2A) and then transferred in the storage solution (Eastwood B, Table 2A) containing 50% glycerol at −20 °C for 1–2 weeks. Then, 20 mM of BDM was added to the skinning and storage solutions.

Skinned trabeculae were clamped with aluminium clips for attachment to the levers of a fast loudspeaker motor and a capacitance force transducer with 50 kHz frequency resonance. A trabecula was then mounted in the experimental chamber between the transducer levers in a drop of relaxing solution. To minimise the end compliance of the skinned trabecula, its extremities were glued to the clips with shellac dissolved in ethanol (Sigma, St. Louis, MO, USA) [34,40]. To recover the distance among myofilaments, increased following skinning, the osmotic agent dextran T-500 (5% *w*/*v*) was added to the experimental solutions [34]. After dextran equilibration, the sarcomere length (SL), width (w), and height (h) of the trabecula were measured at 0.5 mm intervals in the 1.5–2.5 mm central segment of the relaxed trabecula with a 40× dry objective (Zeiss, NA 0.60) and a 25× eyepiece (Objective and eyepiece from Zeiss, Oberkochen, Germany). The trabecula length (*L*_0_) was adjusted to set SL at 2.3 µm.

The cross-sectional area (CSA) of the trabecula was determined assuming the cross-section as elliptical. In the experiments reported here, CSA ranged between 5800 and 19,500 µm^2^. The force exerted by each trabecula was made relative to its CSA and expressed in kPa.

### 4.3. Experimental Protocols

Trabeculae were activated by a temperature jump with the solution exchange system previously described [40] to avoid development of sarcomere inhomogeneities associated with the time taken by Ca^2+^ to diffuse into the myocytes. Trabeculae were allowed to equilibrate in the activating solution (pCa 4.5) at 1 °C for 3–4 s and then transferred to the activating solution at the test temperature (10–30 °C). A striation follower [56] recorded the half-sarcomere length changes from a selected region (600–800 μm long) along the trabecula. During isometric force development, the signal started to be reliable at the time the optic path was permitted through the glass window in the floor of the test temperature drop (see [40] for details).

The dependence of the half-sarcomere stiffness (*k*_0_) and isometric force of maximally activated trabeculae (pCa 4.5, six preparations) on temperature was determined by superimposing small length changes (ranging from –3 to +3 nm per hs, stretch positive), complete in 110 µs, on the steady isometric contraction [33,40]. *k*_0_ was estimated by the slope of the relation between the tension attained at the end of the step and the change in half-sarcomere length (*T*_1_ relation). A train of different-sized steps at 200 ms intervals was applied during each activation in order to enhance the precision of the stiffness measurement. Sarcomere length and isometric tension at the start of each test step were kept constant by applying a step of the same size but in the opposite direction 50 ms after each test step [40].

### 4.4. Solutions

The composition of the solutions, as reported in Table 2, was calculated with a computer program similar to that described by Brandt et al. [57] and Goldman et al. [58], available in the laboratory. Cysteine and cysteine/serine protease inhibitors (*trans*-epoxysuccinil-L leucylamido(4-guanidino)butane, E–64, 10 μM; leupeptin, 20 μg/mL) were also added to all solutions to preserve lattice proteins and thus sarcomere homogeneity. For these experiments, two groups of solutions were prepared: (i) solutions to be used during the experiment (Table 2B)—relaxing solution, containing ATP and no calcium, pre-activating solution, containing HDTA instead of EGTA as a calcium chelator, used to make Ca^2+^ activation faster, and activating solution, containing ATP and calcium (pCa 4.5), and (ii) solutions to be used to prepare and store the trabeculae (Table 2A).

## Figures and Tables

**Figure 1 ijms-26-00469-f001:**
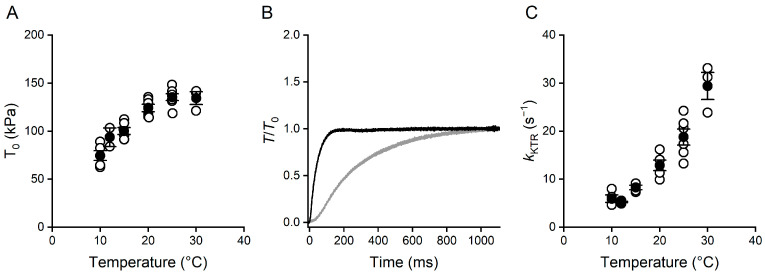
Temperature dependence of isometric force and of the rate of force redevelopment. (**A**) Relation between isometric force at saturating [Ca^2+^], *T*_0_, and temperature. Open symbols = data from 6 trabeculae. Filled symbols = mean ± SEM, n = 2–6. (**B**) Superimposed sample records of force redevelopment following a large release from the steady isometric force at 10 (grey) and 30 °C (black). Force normalised for the steady value reached after the release. (**C**) Relation between rate of force redevelopment, *k*_TR_, measured by the reciprocal of the time to attain 63% of the redeveloped steady force and temperature. Data from the same six trabeculae as in (**A**). Symbols as in (**A**).

**Figure 2 ijms-26-00469-f002:**
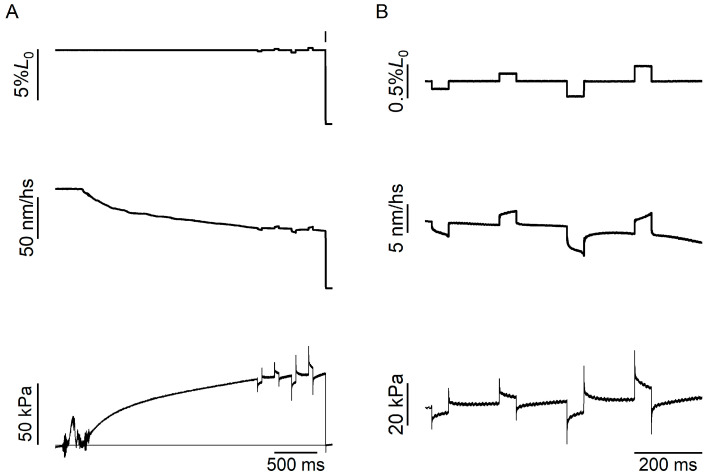
Protocol for measuring the half-sarcomere stiffness during isometric contraction at saturating [Ca^2+^]. (**A**) Force response (lower trace) to a train of length steps (% of the trabecula length, *L*_0_, upper trace), separated by 200 ms, imposed on the trabecula at the plateau of the isometric force developed at pCa 4.5. The horizontal line superimposed on the force response is the force baseline obtained with a large release (~7% of the initial trabecula length), imposed at the time marked by the vertical bar to drop the force to zero. The middle trace shows the half-sarcomere length change (nm/hs) recorded with the striation follower. Note that during the initial force development, there was a shortening of ~50 nm/hs against the end compliances. (**B**) Same traces as in (**A**) on a faster timescale and larger vertical amplification. Each step was followed after 50 ms by a step of the same amplitude in the opposite direction to recover the force and the sarcomere length before the next step. Half-sarcomere shortening/lengthening during the quick force recovery following a step release/stretch were accounted for by the end compliance. Force is expressed relative to the CSA in relaxing solution in the presence of 5% dextran (see the Methods Section). Trabecula length, 2.2 mm; segment length under the striation follower, 0.70 mm; starting sarcomere length, 2.29 μm; test temperature, 10 °C. CSA, 18,200 μm^2^.

**Figure 3 ijms-26-00469-f003:**
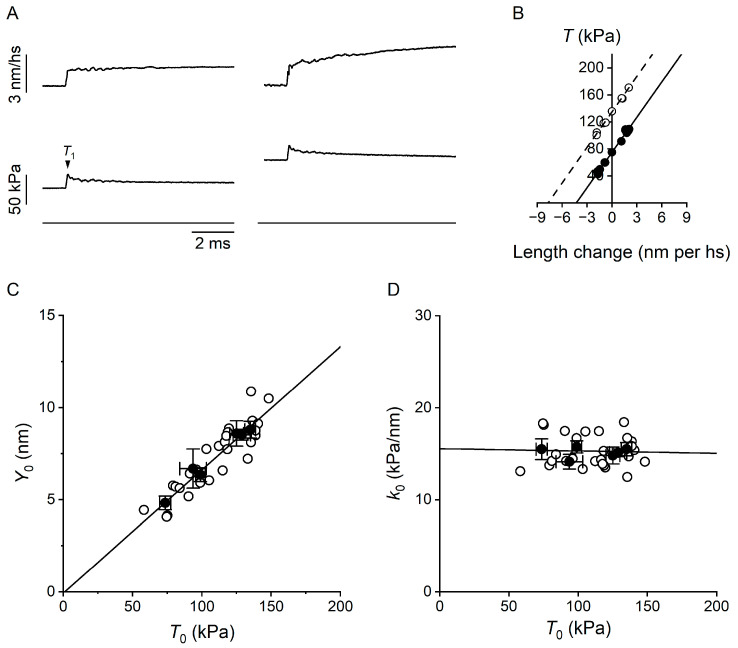
Stiffness and strain of the half-sarcomere in relation to the temperature-dependent changes in the isometric force. (**A**) Force (lower panel, with zero force shown by the bottom horizontal line) and change in the half-sarcomere length (upper panel) in response to a length step imposed on the isometric plateau of a trabecula activated at pCa 4.5, either at 10 °C (left) or at 25 °C (right). *T*_1_, the force attained at the end of the length step, is marked by the arrowhead. (**B**) *T*_1_ relations at 10 °C (filled circles) and at 25 °C (open circles). Solid and dashed lines are first-order regression equations fitted to the data at 10 and 25 °C, respectively. Data from the same trabecula as in Figure 2. (**C**) Relation of *k*_0_ versus the temperature-dependent isometric force. (**D**) Relation of *Y*_0_ versus the temperature-dependent isometric force. In (**C**,**D**): open circles, data from 6 trabeculae; filled circles, mean ± SEM (n = 2–6).

**Figure 4 ijms-26-00469-f004:**
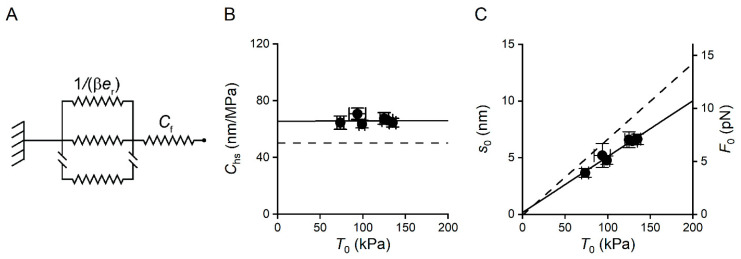
Mechanical model of the hs and dependence of half-sarcomere compliance (*C*_hs_) and the average strain of the attached motors (*s*_0_) on temperature-modulated *T*_0_. (**A**) Two-component model. Parameters identifying the compliance of the array of motors (1/(β·*e*_r_) and the equivalent compliance of the filaments (*C*_f_) defined in the text. (**B**) Dependence of *C*_hs_ on *T*_0_. The continuous line is the first-order regression to the data points (filled symbols), calculated from the averaged data in Figure 3C. The slope is not significantly different from zero (*p* = 0.96), and the ordinate intercept is 65 ± 6 nm/MPa. The dashed line shows the contribution of *C*_f_ to *C*_hs_. (**C**) Dependence of *s*_0_ (left ordinate, calculated as detailed in the text) and the force per motor (*F*_0_; right ordinate, calculated as detailed in the text) on *T*_0_. The continuous line is the linear fit to data points and has an intercept not significantly different from the origin (*p* = 0.8). The dashed line shows the hs strain, from Figure 3D. Data are mean ± SEM from six trabeculae.

**Figure 5 ijms-26-00469-f005:**
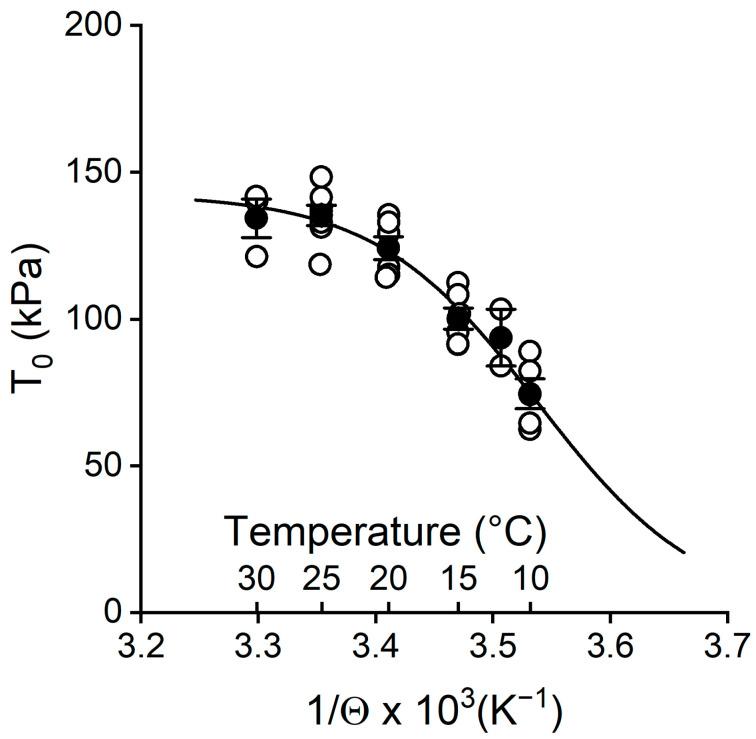
Relation between steady isometric force developed at pCa 4.5 and the reciprocal of absolute temperature Θ. Circles are mean ± SEM from Figure 1A. The line is the fit to the pooled data using (4).

**Table 1 ijms-26-00469-t001:** Dependence of the relevant mechanical and kinetic parameters on temperature. Data are mean ± SEM from 6 trabeculae. One-way analysis of variance (ANOVA) demonstrated that temperature did not significantly affect hs stiffness, *k*_0_, (*p* = 0.89), while it significantly affected *T*_0_ and *k*_TR_ (*p* < 0.0001).

	10 °C	15 °C	20 °C	25 °C	30 °C	Average Q_10_ (10–25 °C)
*T*_0_ (kPa)	74 ± 4	99 ± 3	125 ± 5	135 ± 4	129 ± 6	1.5
*k*_TR_ (s^−1^)	5.9 ± 0.8	8.3 ± 0.5	12.9 ± 1.1	18.8 ± 1.7	29.3 ± 2.8	2.2
*k*_0_ (kPa nm^−1^)	15.5 ± 1.1	15.7 ± 0.7	14.8 ± 0.9	15.5 ± 0.7	15.1 ± 0.4	

**Table 2 ijms-26-00469-t002:** Composition of the solutions. (**A**) Solutions used to prepare and store demembranated trabeculae: Eastwood A, skinning solution, and Eastwood B, storage solution. (**B**) Solutions used during the experiment. All concentrations are in mM, except glycerol (% *v*/*v*). ATP, adenosine 5′-triphosphate; EGTA, ethylene glycol-bis (β-aminoethyl ether)-N,N,N′,N′-tetraacetic acid; HDTA, 1,6 diaminohexane-N,N,N′,N′-tetraacetic acid; TES, N tris[hydroxymethyl]methyl-2-aminoethanesulphonic acid; Na_2_CP, phosphocreatine disodium salt hydrate; GSH, glutathione; KP, potassium propionate; PMSF, phenylmethylsulphonyl fluoride. Here, 1 mg/mL of creatine phosphokinase, 10 µM of *trans*-epoxysuccinyl-L-leucylamido(4-guanidino) butane (E-64), and 20 µg/mL of leupeptin were added to all solutions. The pH was adjusted with KOH or HCl to 7.1. Because of the temperature dependence of the pK of the buffer used, to keep the ionic strength constant at ~190 mM independent of the solution temperature, the concentration of TES was decreased as the temperature increased (from 112 mM at 10 °C to 50 mM at 30 °C). Pre-activating solution at 5 °C: TES, 140 mM. Free Mg^2+^ was in the range 1.7–1.9 mM and MgATP was in the range 4.9–5 mM. All chemicals were obtained from Sigma (St. Louis, MO, USA). Five percent (*m*/*v*) of dextran T500 (Thermo Fisher Alfa Aesar, Haverhill, MA, USA) was added to all solutions in (**B**), and 20 mM of BDM was added to the skinning and storage solutions.

A								
	Imidazole	MgCl_2_	Na_2_ATP	EGTA	KP	PMSF	GSH	Gly
Eastwood A	10	2.5	2.5	5	170	0.2	−	−
Eastwood B	20	5	5	10	340	−	20	50
**B**								
	**TES**	**MgCl_2_**	**Na_2_ATP**	**EGTA**	**CaEGTA**	**Na_2_CP**	**GSH**	**HDTA**
Pre-Activating (5 °C)	140	6.93	5.45	0.1	−	19.49	10	24.9
Relaxing (10 °C)	112	8.40	5.44	25	−	19.11	10	−
Activating (10 °C)	112	6.76	5.46	−	25	19.49	10	−

## Data Availability

The original contributions presented in this study are included in the article. Further inquiries can be directed to the corresponding author.

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
