# Peer review of "The Mechanism of Modulation of Cardiac Force by Temperature"

_ijms, 2025, doi:10.3390/ijms26020469_

Round 1

Reviewer 1 Report

Comments and Suggestions for Authors

Dear Editor,

Thank you for asking me to review this manuscript titled "The Mechanism of Modulation of Cardiac Force by Temperature" by Dr.  Morotti and colleagues.

This is an interesting experimental study where the authors utilize sarcomere - level mechanics to investigate the effect of low temperature on cardiac muscle contraction and performance.

The experimental design and setup appear sound and well developed.

I have the following comments:

COMMENTS:

1. The introduction is quite extensive and verbose!! It is understandable one wishing to be thorough but much of the information provided is more appropriate for the discussion section. It needs to be trimmed down and made more focused!

2. How many repetitions of measurements was performed by each extracted heart? How many bundles were removed from each and were data pooled together or best values used?

3. Why wasn't statistical analysis performed between different temperatures to demonstrate the significant decline in function/contraction force/stiffness?

4. Where the principles of 4 Rs followed in this experiment? It is not mentioned!

In conclusion, this is an interesting study with solid outcomes reported which has scientific significance and value and could potentially be reported following some minor editing.

Reviewer 2 Report

Comments and Suggestions for Authors

1. Provide a more comprehensive comparison with previous studies, particularly those reporting the lack of temperature effects on force per motor in cardiac trabeculae, to contextualize the findings.

2. Expand on the molecular basis of the endothermic force-generating transition in cardiac myosin. Explain how the interaction of hydrophobic surfaces relates to observed changes in force per motor.

3. Provide a clearer explanation of the simplified mechanical model used to interpret the data. Include assumptions, limitations, and why it is suitable for this analysis.

4. Use consistent terminology throughout the manuscript (e.g., "force per motor," "ensemble force," "myosin motor force") to improve clarity.

5. Suggest future research directions, such as the exploration of myosin motor modifications or interactions at different temperature ranges or in pathological conditions.
